# Linking the Autotaxin-LPA Axis to Medicinal Cannabis and the Endocannabinoid System

**DOI:** 10.3390/ijms25063212

**Published:** 2024-03-12

**Authors:** Mathias C. Eymery, Ahcène Boumendjel, Andrew A. McCarthy, Jens Hausmann

**Affiliations:** 1European Molecular Biology Laboratory, EMBL Grenoble, 71 Avenue des Martyrs, 38000 Grenoble, France; meymery@embl.fr; 2Laboratoire Radiopharmaceutiques Biocliniques, INSERM U1039, Université Grenoble Alpes, 38000 Grenoble, France; ahcene.boumendjel@univ-grenoble-alpes.fr; 3School of Medicine and Health Sciences, Carl von Ossietzky University of Oldenburg, 26129 Oldenburg, Germany

**Keywords:** autotaxin, lysophosphatidic acid, cannabinoids, tetrahydrocannabinol, endocannabinoid

## Abstract

Over the past few decades, many current uses for cannabinoids have been described, ranging from controlling epilepsy to neuropathic pain and anxiety treatment. Medicines containing cannabinoids have been approved by both the FDA and the EMA for the control of specific diseases for which there are few alternatives. However, the molecular-level mechanism of action of cannabinoids is still poorly understood. Recently, cannabinoids have been shown to interact with autotaxin (ATX), a secreted lysophospholipase D enzyme responsible for catalyzing lysophosphatidylcholine (LPC) to lysophosphatidic acid (LPA), a pleiotropic growth factor that interacts with LPA receptors. In addition, a high-resolution structure of ATX in complex with THC has recently been published, accompanied by biochemical studies investigating this interaction. Due to their LPA-like structure, endocannabinoids have been shown to interact with ATX in a less potent manner. This finding opens new areas of research regarding cannabinoids and endocannabinoids, as it could establish the effect of these compounds at the molecular level, particularly in relation to inflammation, which cannot be explained by the interaction with CB1 and CB2 receptors alone. Further research is needed to elucidate the mechanism behind the interaction between cannabinoids and endocannabinoids in humans and to fully explore the therapeutic potential of such approaches.

## 1. Introduction

### 1.1. Introduction to Medicinal Cannabis and the Endocannabinoid System

Medicinal cannabis is commonly used in many countries despite restrictive laws prohibiting its use. It is also referred to as therapeutic cannabis or medicinal marijuana, while its botanic name is *Cannabis sativa*. Recent interest by patients and healthcare providers made cannabis use more tolerated in many countries such as Belgium, the Netherlands, the United Kingdom, Canada, Spain, and several states in the USA. The use of such therapeutic approaches is restricted to patients with a prescription or a specific diagnosis and distributed under the responsibility of a pharmacist. One of the limitations of medicinal cannabis is the composition of the plant extract used, which often contains a large number of molecules that are classified as “cannabinoids”, sharing common structural features [1]. The most studied cannabinoid is delta-9-tetrahydrocannabinol (or for simplicity referred to from here as THC) and its structure was discovered by Mechoulam *et al*. in 1964, following isolation from hashish [2]. A close derivative is delta-8-terohydrocannabinol (delta-8-THC), in which only the position of unsaturation, 6a–10a, on the C ring, differs, as shown in Figure 1, where the THC molecule is shown in red. Delta-9-tetrahydrocannabinolic acid (THCA) corresponds to the derivative produced naturally by the plant, which is further decarboxylated and activated by heating before consumption or smoking to lead to THC. Another isomer of THC is cannabinol (CBN), which differs by the presence of an aromatic ring instead of a single unsaturation in the C ring. This derivative is also less active, probably due to the higher rigidity of the molecule. Another important class of derivative is represented by cannabidiol (CBD), again less potent than THC due to the opening of the B ring. Other derivatives will not be described here but they are widely studied and an abundance of literature can be found [2,3].

Later, Raphael Mechoulam also discovered other molecules in the human body, named “endocannabinoids”, because they were described as the endogenous ligands for the cannabinoid receptor 1 and 2 (CB1 and CB2) [4,5]. CB1 and CB2 bind cannabinoids and endocannabinoids with varying affinity. The CB receptors were discovered more than 40 years ago during in vitro studies and belong to the large GPCR family [6]. CB1 is one of the most expressed receptors in the brain and is responsible for retrograde signaling when bound to any agonist endocannabinoid. This receptor can induce a depolarization of neurons that will reduce GABA-mediated neurotransmission. CB2 is expressed in the immune system, the central nervous system, and the brain. This receptor is involved in nociception and is responsible for pain relief. The CB2 receptor is probably involved in many other cellular processes that remain unknown [6,7].

Anandamide (AEA) [8] and 2-arachidonoylglycerol (2-AG) [9], which were detected in samples from the brain and intestine and shown to activate CB1 and CB2 receptors with high affinity and efficacy, are the best characterized endocannabinoids [10]. Six molecules have been linked to the endocannabinoid human system to date, namely AEA, 2-AG, 2-arachidonyl glyceryl ether (2-AEA), N-Arachidonoyl dopamine (NADA), lysophosphatidylinositol, and virodhamine. Their chemical structures are displayed in Figure 2 and we can notice some structural similitudes to LPA as they all contain a long lipophilic tail.

More recently, enzymes for the biosynthesis and the degradation of endocannabinoids have been found, in particular N-acetylphosphatidylethanolamine-specific phospholipase D (NAPE-PLD) that is responsible for catalyzing the synthesis of anandamide and other N-acylethanolamines [11]. The fatty acid amine hydrolase (FAAH) is able to catalyze the degradation of AEA, other N-acylethanolamines, and fatty acid-derived primary amides [12]. Diacylglycerol lipase α (DAGLα) and DAGLβ are involved in the synthesis of 2-AG as well as other monoacylglycerols [13]; monoacylglycerol lipase (MAGL) catalyzes the breakdown of 2-AG and other monoacylglycerols [14]. Moreover, it has been shown that the enzymatic dephosphorylation of a 2-arachidonoyl species of lysophosphatidic acid (LPA) in the brain of rats generates the formation of 2-AG [15], a process that has been shown to depend on lipid phosphate phosphatases [16]. These findings expand our knowledge about what is better described as the endocannabinoid system [3]. The endocannabinoid system is part of the endocannabinoidome, a much wider and more complex network of promiscuous mediators overlapping with other signaling pathways [3] that are involved in many physiological processes through the fine-tuning of various GPCR signals.

Many G-protein-coupled receptors (GPCRs) have been characterized during the last few decades, with many being important for physiological and pathological responses to external stimuli. Prominent among these are the six lysophosphatidic acid receptors (LPA_1–6_), binding LPAs produced by ATX and other enzymes, and cannabinoid (CB) receptors, which bind endocannabinoids. LPA and CB receptors are both involved in mediating physiological processes, in particular in the context of pain and inflammation. Recently, the crosstalk between LPA_1_ receptors and CB1 has been demonstrated by Gonzàlez de San Román *et al*., showing that the absence of one or the other is able to modulate the other system at the signaling level and also at the neurotransmitter level during prenatal or postnatal development in mice [17]. Sordelli *et al*. also showed that in the rat uterus, the LPA_3_ receptor participates in the implantation process through interaction with other lipid derivatives, in particular prostaglandins and endocannabinoids [18]. From a chemical point of view, the link between LPA and endocannabinoids could be explained partially by the structural similarity they share: a long lipophilic tail with a head group. Nakane *et al*. found that 2-arachidonoyl LPA can be converted to 2-AG in a brain homogenate, unraveling that two different endogenous ligands that interact with specific receptors can be related metabolically and interconverted with fast kinetics [15]. The structural comparison of the more common LPA 20:4 and 2-AG shows many similarities (Figure 3).

### 1.2. Autotaxin: A Potential Drug Target

Autotaxin is a 100–130 kDa secreted protein that is folded into four domains, namely the somatomedin B-like domains (SMB 1 and 2), the catalytic phosphodiesterase domain (PDE), and the inactive nuclease domain (Nuc). Three major isoforms, ATX-α, ATX-β, and ATX-γ, have been identified, where ATX-β is the canonical plasma isoform and ATX-γ is the neuronal-specific one. All ATX isoforms contain a bimetallo Zn catalytic site and multiple glycosylated residues that are important for their catalytic activity. ATX is the main protein responsible for the production of bioactive LPA from LPC, an important mediator that can bind to six different G-protein coupled receptor subtypes, LPA_1–6_ [1,19]. Many functions for LPA receptors have been described, ranging from cell homeostasis to malignant cell development [20]. Recently, it has been highlighted that ATX also acts as a chaperone, presenting LPA to its cognate receptor and reassessing the importance of the microenvironment for the ATX-LPA signaling axis [21,22]. This new finding and the interaction of ATX with other cell surface receptors such as integrins helps explain how signaling at the cell surface can occur independently of the LPA blood level. From a chemical point of view, it is also important to mention that LPA molecules comprise a large number of derivatives, with different lipophilic tails but with all sharing the same phosphate head group. As mentioned previously, the lipophilic part of LPAs shares structural similarities with endocannabinoids.

It is now well established that ATX is a major contributor to several diseases, making ATX an interesting drug target, as shown by the increasing number of inhibitors developed during the last decade [23,24,25,26,27]. ATX has been linked to tumor progression in melanoma cells and this effect can be explained by the importance of LPA to stimulate cell migration and division. These ATX and LPA effects were also reported in other studies, such as ATX overexpression in small lung cancers, ovarian cancer, and glioblastoma. All these studies highlight ATX as a potential target for cancer, with screening for overexpression at tumor sites in line with personalized medicine therapeutic strategies.

ATX has also been implicated in many other diseases, including platelet aggregation, neuropathic pain, stroke, acute coronary syndrome, bone development, and vascular homeostasis [28,29,30,31,32]. Recent studies showed how the ATX-LPA signaling axis is important for blood vessel formation in zebrafish and mice models [33,34]. It has further been shown that ATX is important for lymphoid trafficking and the migration of lymphocytes to lymphoid organs, which might explain some of the effects of ATX on inflammation [35]. Adipocytes are the cells that primarily express ATX, producing more than 50% of LPA levels, linking the ATX-LPA axis to obesity [36,37] and subsequent related diseases such as insulin resistance [36,37] and chronic inflammation [33,35,38]. It is now widely accepted that ATX is linked to cardiovascular diseases [30] as it has been demonstrated that LPA can activate platelets [39], a leading cause of stroke and atherosclerosis in pathologic conditions [40]. The effect of LPA on endothelial cells, not only during angiogenesis, as mentioned before, but in adhesive molecule expression and vasodilatation has also been reported [41]. Lastly, it is important to mention that ATX has been implicated in idiopathic pulmonary fibrosis, with high levels of ATX and LPA found in the bronchopulmonary fluids of affected patients [42]. This can lead to multiple activations of LPA receptors, resulting in hypersecretion of proinflammatory mediators as well as accumulation of fibroblasts [23,33].

More recent research linked ATX to neurological diseases, with the detection and the characterization of the ATX-γ isoform in the central nervous system [43,44]. LPA has been widely studied for its pleiotropic effect but also for its specific role in triggering neurological diseases, such as neuropathic pain, Alzheimer’s disease, glioblastoma multiform, schizophrenia, multiple sclerosis, and traumatic brain injuries [45]. Neuropathic pain has been linked to ATX and LPA levels, with higher pain intensity in individuals with increased LPA levels [46]. This has also been demonstrated in vitro by Inoue *et al*., who showed that LPA_1_-deficient mice did not develop neuropathic pain after nerve injury [47]. This could be partially explained by the crosstalk between LPA and CB receptors, which could mediate neuropathic pain symptoms. Concerning other pathological conditions such as Alzheimer’s disease, it has been shown that ATX expression and activity dysregulation were linked to this disease, paving the way for new therapies targeting this deadly illness [48].

### 1.3. Cannabinoids and Endocannabinoids: Interaction with Human Proteins

Both CB1 and CB2 receptor structures were recently deciphered (Figure 4) [49,50]. These structures provide a basis for the structure and function of the CB1 receptor when bound to cannabinoids and are available from the PDB database (PDB ID: 5U09 and 5TGZ). The CB2 receptor structure was also deciphered (PDB: 5ZTY) [51], providing molecular insights on the difference between CB1 and CB2 receptors, leading to a characterization of differences and the distinct antagonist-binding mode of the CB2 receptor. CB1 and CB2 receptors have a 44% sequence homology but interestingly, their ligand-binding pockets are structurally conserved orthostatic sites, as described below. This explains the binding of ligands with structural similarities and the failure to develop a selective inhibitor. Zhi-jie Liu *et al*. have worked extensively on CB1 and CB2 receptor structural determination. They deciphered the structure of CB receptors in various states and it is now recognized that CB receptors have at least three different functional states: an antagonistic, an intermediate, and an activated state that mediates downstream signaling of the receptors [52].

The structure of endocannabinoids and cannabinoids bound to various human proteins such as fatty acid binding protein 5 (FABP5), peroxisome proliferation activating receptor gamma (PPAR-γ), and CB1/CB2 receptors [53,54,55,56] are available in the PDB. These recent discoveries have been made possible due to technological improvements in Cryo-EM and macromolecular crystallography. Recently, Yang *et al*. compared the binding site of CB1 and CB2 receptors [57]. They both share a hydrophobic pocket, which is important to bind the highly hydrophobic cannabinoid and endocannabinoid molecules. The hydrophobic interactions are mediated by tryptophan and phenylalanine, which are conserved between CB1 and CB2 receptors. A histidine facilitating a hydrogen bond between the ligands and receptors is also present in both CB1 and CB2 receptors.

Recently, PPAR-γ binding to cannabinoids has been shown to mediate neuroprotection, reward, memory, cognition, and analgesic effects. Furthermore, it has been demonstrated that PPAR-γ can trigger various processes such as apoptosis during cannabinoid treatment of cancer cell lines. Effects on inflammation, satiety, metabolism, and vasorelaxation have also been reported [53].

FABP5 is an intracellular chaperone of fatty acid molecules that regulates lipid metabolism and cell growth. FABP5 has recently been reported to bind cannabinoids and endocannabinoids. It has further been shown that this protein interacts with the endocannabinoid system as a carrier and is indispensable for their transport at central glutamate synapses [54].

## 2. Discussion

### 2.1. Recent Advances in Medicinal Cannabis Therapy

Many clinical trials involving cannabinoids and endocannabinoids have been conducted for a large number of diseases such as refractive epilepsy, neuropathic pain, cancer, multiple sclerosis, anxiety, nausea, and anorexia. Despite the huge number of clinical trials conducted, there has been no clear result as to whether this therapy is as good or better than already approved therapeutics. The legislation on medicinal cannabis is still very restrictive in most countries and the difficulty in patenting cannabinoids is probably a limitation for clinical trial investment from the private sector. Most of the clinical trials are from public agencies with a small number of patients, with the aim of proposing alternative therapies to patients with severe diseases. However, the pharmacokinetics of THC and CBD are well studied and data are available assessing their safety [58]. More clinical trials are clearly needed to determine whether cannabinoid therapy is providing sufficient improvement compared to the side effects of their administration. Some precautions should be taken since studies have also shown that THC permanently affects brain development, especially when used at a young age since a study highlighted that marijuana was associated with a loss in IQ points if used during adolescence [59,60]. Adults do not appear to be concerned by IQ decline if they commenced consumption after brain development was complete. Other side effects are reported in the literature, such as breathing impairments and lung cancers. This can be avoided by consumption of oral forms of cannabinoids, such as oil extracts. It has also been reported that cannabis increases the heart rate for several hours after intake, with a concomitant increased risk of heart attack. It is also dangerous for pregnant women to consume cannabis since its use has been linked with lower birth weight and a higher risk of brain malformations and other neurological impairments [61]. All this underscores the need to better understand the interaction of medicinal cannabis constituents with pharmacological targets in order to assess the therapeutic value of such treatments.

### 2.2. Inhibition of Autotaxin by Cannabinoids

THC works as a partial inhibitor on the catalysis of both ATX-β and ATX-γ isoforms, as described previously by Eymery *et al*. [62]. The apparent EC_50_ of THC with ATX-β and the substrate LPC 18:1 is 1025 ± 138 nM, with a magnitude of inhibition around 60%. For ATX-γ, the apparent EC_50_ is 407 ± 67 nM for THC, with a similar magnitude of inhibition [62]. To decipher the binding interface, autotaxin was co-crystallized with THC [62] (PDB ID: 7P4J). THC binds rat ATX-β in the hydrophobic pocket with the aliphatic chain pointing into the end of this pocket. THC binding is dependent on many hydrophobic interactions, with the residues I167, F210, L213, L216, W254, F274, Y306, and V365 all contributing. A superposition of the rATX-β-THC structure with rATX-β-LPA 18:1 (PDB ID: 5DLW) [63] clearly shows that the THC molecule blocks the binding of the LPA 18:1 aliphatic chain. However, the glycerol backbone and phosphate group of LPA can still bind to the active site, according to the structural data described by Eymery *et al*. [62].

### 2.3. Inhibition of Autotaxin by Endocannabinoids

Structural similitudes between 2-AG and 20:4 LPA suggested that this endocannabinoid might interact with ATX and interfere with its catalytic function. Indeed, it has been reported that 2-AG and AEA can inhibit ATX when tested with both LPC16:0 and LPC 18:1 as substrates. A partial inhibition of ATX-β has been observed with both endocannabinoids when using a low LPC concentration of 20 µM [1], with a reported apparent EC_50_ value for 2-AG of 4.1 ± 1.3 µM with LPC 16:0 and 10.6 ± 2.2 µM with LPC 18:1 as substrates, respectively. AEA was reported to have a weaker inhibition with an apparent EC_50_ of 8.1 ± 2.3 µM with LPC 16:0 and 18.6 ± 4.5 µM with LPC 18:1 as substrates, respectively. 2-AG is reported to have a maximal percentage of inhibition of 65–70% for both substrates and AEA a maximal percentage of inhibition of 55% for LPC 16:0 and 75% for LPC 18:1. However, when the experiment was repeated with a more physiological LPC concentration of 200 µM found in serum, little or no inhibition could be detected.

## 3. Future Directions

### 3.1. Investigate the Effect of Medicinal Cannabis and Endocannabinoids on the ATX-LPA Axis

As previously mentioned, medicinal cannabis can affect various pathologies without a clear indication of their mode of action. Ideally, to show if medicinal cannabis, and particularly THC, has an effect on whole blood LPA concentration, a comparison of patient LPA levels before and after starting a treatment involving cannabinoids could be considered. It is feasible to measure LPA levels by mass spectrometry and to compare these results [38]. LPA levels are not very stable and are known to depend on many factors. It would therefore be important to exclude patients presenting metabolic disorders or those overweight from such studies [64]. Another possibility is to perform a simple pharmacokinetics study by measuring LPA levels at different time points after the administration of cannabinoids over several hours. This kind of study could also be performed in animals with cannabinoids. It is also of great interest to measure LPA levels in patients treated with medicinal cannabis to assess if the inhibition of ATX is significant in vivo and if it remains over a longer period. As some of the therapeutic effects of THC might come from the inhibition of ATX, more experiments are needed to investigate this option. For example, it would be interesting to monitor the number of crises an epilepsy animal model presenting an inducible ATX knockout undergoes and verify if the number of crises is lowered when ATX is knocked out.

### 3.2. Development of New ATX Inhibitors Inspired by Cannabinoids and Endocannabinoids

Many ATX inhibitors have been developed over the last few decades and several have entered clinical trials but none have passed the regulatory controls to reach the market to date. For example, the inhibitors developed by Galapagos failed in Phase 2/3 trials due to their potential toxicity in patients with idiopathic pulmonary fibrosis. This is likely due to the various biological effects of ATX on the body and highlights the necessity to target the inhibition of ATX at the disease site rather than the whole body. The pharmacokinetics of the inhibitor might also be an issue due to the high turnover of ATX in the body [23,65]. Indeed, inhibitors might need to be taken several times a day to maintain ATX inhibition and subsequently reduce LPA production. The development of cannabinoid-inspired ATX inhibitors might be interesting from this point of view since their lipidic structure allows potential deposits in fatty tissues and lipid carrier proteins to prolong the limited inhibition following their administration [44,66,67].

## 4. Conclusions

In conclusion, the review of recent studies shown here warrants further research into the pleiotropic effects of medicinal cannabis and endocannabinoids in the context of ATX-LPA signaling, while also providing a promising starting point for future research directions. Moreover, the crosstalk between LPA and CB receptors is of medical interest, especially when considered in the context of medicinal cannabis use and ATX inhibition by cannabinoids and endocannabinoids.

## Figures and Tables

**Figure 1 ijms-25-03212-f001:**
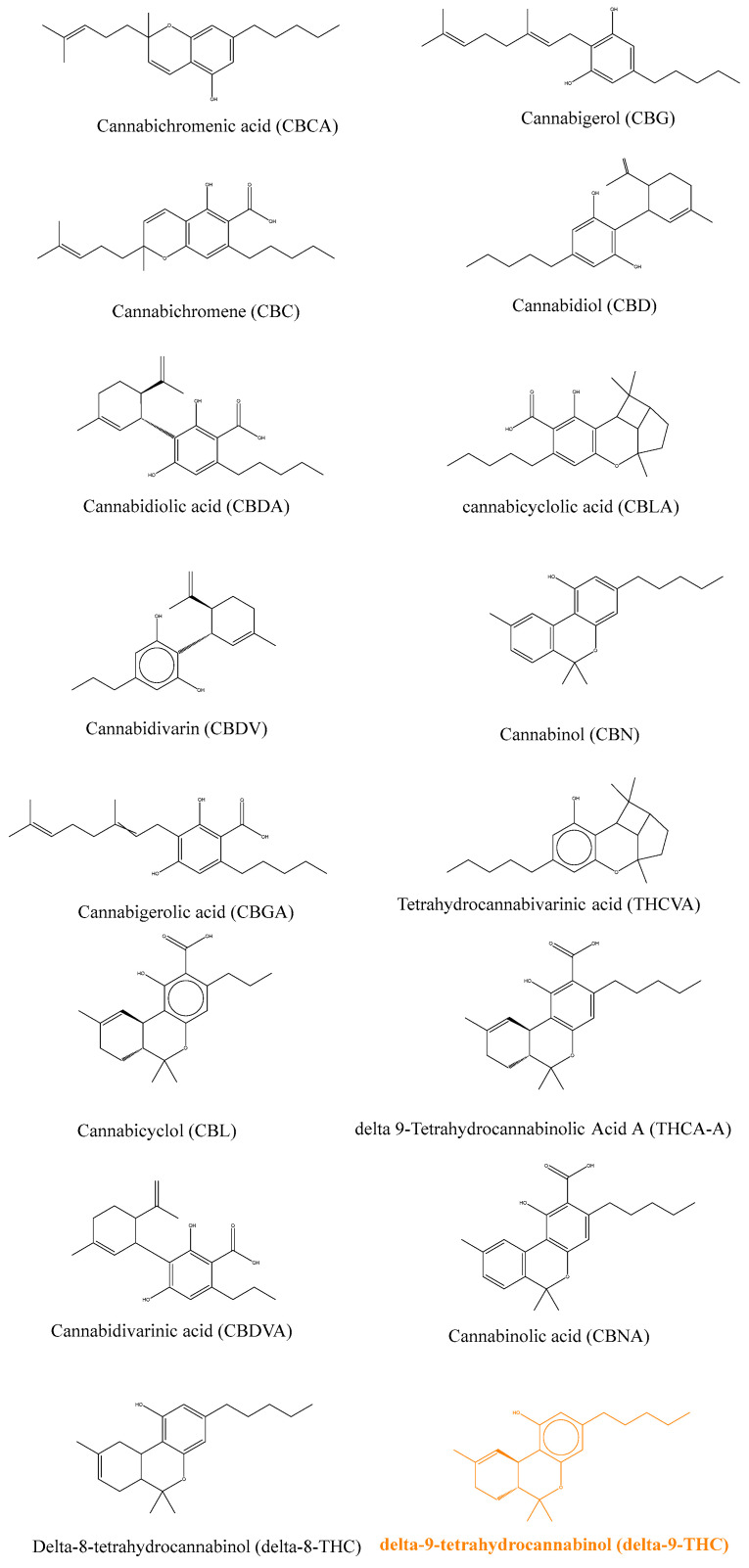
Chemical structures of the main plant cannabinoids [1].

**Figure 2 ijms-25-03212-f002:**
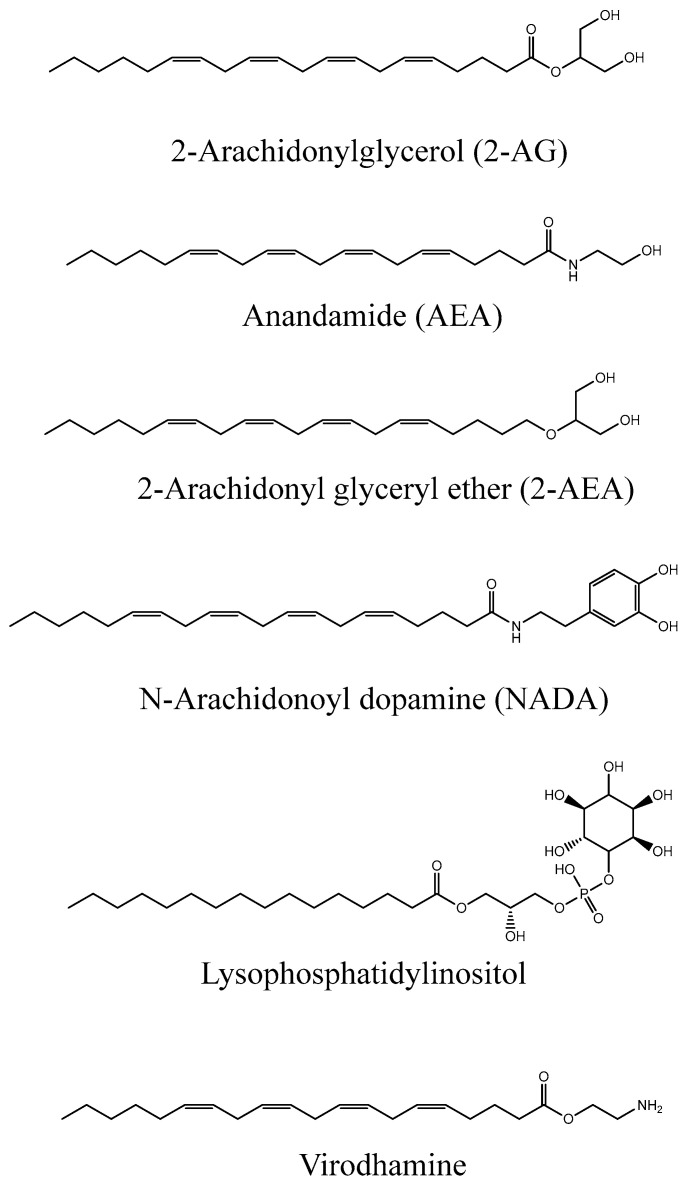
Chemical structures of human endocannabinoids [1].

**Figure 3 ijms-25-03212-f003:**
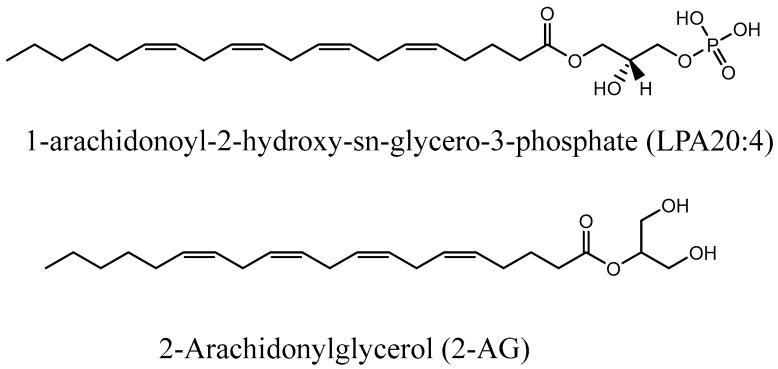
Structural comparison of LPA 20:4 and 2-AG.

**Figure 4 ijms-25-03212-f004:**
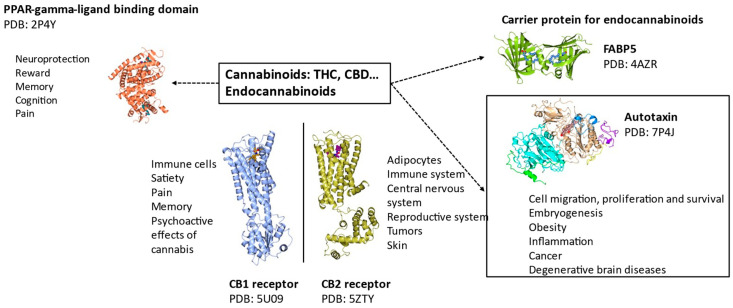
Structure–function of human proteins interacting with cannabinoids and endocannabinoids [1].

## Data Availability

Not applicable.

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
