# Peer review of "Linking the Autotaxin-LPA Axis to Medicinal Cannabis and the Endocannabinoid System"

_ijms, 2024, doi:10.3390/ijms25063212_

Round 1

Reviewer 1 Report

Comments and Suggestions for Authors

Dear Authors! 

I have read your manuscript, and my comments are given below. 

First, while the abstract is well written and understandable for the reader, this is not the case with the introduction. It should introduce the theme and should not begin with your aim of review. Further on, the second paragraph is confusing. If one does not read the abstract, it is hard to understand what you are trying to explain. Please rewrite the beginning of the introduction to make it clearer to the readers. 

The chapter 1.1 is too long; why explaining all the history of the cannabis usage? Especially the 2nd paragraph is redundant (page 2, line59-73). 

On page 3, line 86 please explain the ANSM abbreviation for the non-European readers. 

I like chapters 1.2. and 1.3. No need to change them. 

In the Discussion, on page 9, chapter 2.3 are those your unpublished results? It is unusual to write of such results in review article is it not? I also searched the 14th reference and could not find it. In the References section only published material should be written. 

I have no comments on the last two chapters, namely Future directions and Conclusions. 

Author Response

We thank reviewer #1 for their time to carefully review our submitted manuscript. You can find below a point-by-point response to your comments, which we hope will address your major concerns.

First, while the abstract is well written and understandable for the reader, this is not the case with the introduction. It should introduce the theme and should not begin with your aim of review. Further on, the second paragraph is confusing. If one does not read the abstract, it is hard to understand what you are trying to explain. Please rewrite the beginning of the introduction to make it clearer to the readers. 

The authors are very grateful for the reviewer’s comment. We have taken this feedback and significantly re-wrote the beginning of the introduction for better readability (see track changes) and flow. We believe these changes have improved the manuscript and again thank the reviewer.

The chapter 1.1 is too long; why explaining all the history of the cannabis usage? Especially the 2nd paragraph is redundant (page 2, line59-73). 

See above, we agree and following the reviewer’s advice we have removed most of this history as it does not add to the scientific relevance of the review.

On page 3, line 86 please explain the ANSM abbreviation for the non-European readers. 

This abbreviation has now been removed.

I like chapters 1.2. and 1.3. No need to change them. 

In the Discussion, on page 9, chapter 2.3 are those your unpublished results? It is unusual to write of such results in review article is it not? I also searched the 14th reference and could not find it. In the References section only published material should be written.

The authors are very grateful for the reviewer’s comment. We agree it’s unusual to present unpublished results in a review so we have therefore removed Figure 5. We have kept section 2.3 and updated the reference to these experimental results, the thesis of Mathias Eymery, which has just been published online. We believe these are important to highlight as the LPC concentration in cerebrospinal fluid is likely to be significantly lower than that found in serum.

I have no comments on the last two chapters, namely Future directions and Conclusions. 

Reviewer 2 Report

Comments and Suggestions for Authors The review presented by Eymery et al., can be a valuable source of information about possible link between the protein autotaxin and the endocannabinoid system. The paper is well structured, the figures complement the text well, the reference list covers the relevant literature adequately (there are many references to recent studies).  The manuscript informatively describes the research background for Cannabinoid- and Autotaxin- related studies. Moreover, the authors consider the use of medicinal cannabis  in the context of  Autotaxin inhibition by cannabinoids and endocannabinoids.

Author Response

We thank reviewer #2 for their time to carefully review our submitted manuscript.
